# Photothermal Therapy Combined with Chemotherapy and Anti-Inflammation Therapy Weakens the Immunosuppression of Cervical Cancer

**DOI:** 10.3390/ph18111657

**Published:** 2025-11-01

**Authors:** Xiaojing Yang, Jie Fu, Yi Xu, Dejian Li, Hanru Ren

**Affiliations:** 1Department of Radiation Oncology, Shanghai Sixth People’s Hospital Affiliated to Shanghai Jiao Tong University School of Medicine, Shanghai 200233, China; youngshanghai@126.com (X.Y.); fujie74@sjtu.edu.cn (J.F.); 2Division of Urology, Department of Surgery, Brigham and Women’s Hospital & Harvard Medical School, Boston, MA 02115, USA; 3Department of Oncology, Shanghai Sixth People’s Hospital Affiliated to Shanghai Jiao Tong University School of Medicine, Shanghai 200233, China; xuyii420@sjtu.edu.cn; 4Department of Orthopedics, Shanghai Pudong Hospital, Pudong Medical Center, Fudan University, Shanghai 201339, China

**Keywords:** photothermal therapy, anti-inflammation therapy, inflammation tumor microenvironment, immunosuppression

## Abstract

**Background/Objectives:** A non-toxic nano-platform which can increase drug-loading rate and synergistically increase antitumor effect is very ideal. This study provides the concept that a combination of photothermal therapy with chemotherapy and anti-inflammatory therapy will be achieved by ablation of the local tumor, robust strategies for the suppression of distant tumors with enhanced antitumor therapy outcomes. **Methods:** In this study, the chemotherapeutic drug cisplatin (DDP) and the anti-inflammatory drug Aspirin-DL-Lysine (ADL) were loaded into a hollow porous nanomaterial zeolitic imidazolate framework-8 (ZIF-8), which was then coated with polydopamine, in order to form near-infrared absorption organic nanoparticles DDP-ADL@ZIF-8@PDA with excellent photothermal conversion efficiency. The antitumor efficacy of the nanodrug was evaluated through physicochemical characterization, cell biology studies, and animal experiments. **Results:** Photothermal therapy (PTT) of polydopamine combined with DDP and ADL can reduce inflammation and the immunosuppressive tumor microenvironment, and enhance antitumor effect. The results showed that the combined therapy could effectively eliminate the primary tumor, shrink the distant tumor, and inhibit the metastasis of the tumor. PTT in combination with chemotherapy and anti-inflammatory therapy can inhibit the expression of inflammatory factors, significantly reducing tumor immunosuppression by eliminating bone marrow-derived suppressor cells and increasing levels of cytotoxic T lymphocyte. **Conclusions:** This study successfully developed a DDP-ADL@ZIF-8@PDA nanomedicine for effective drug delivery, synergistic photothermal therapy, and anti-inflammatory attenuated immunotherapy to enhance treatment of human cervical cancer xenografts in mice. Overall, the combination of photothermal therapy with chemotherapy and anti-inflammatory therapy on a nano-platform has great potential for antitumor therapy applications.

## 1. Introduction

Cervical cancer is a common female malignant tumor [1]. Advanced and recurrent cervical cancer is fatal and often requires chemotherapy [2]. Traditional chemotherapeutic drugs have obvious toxic and side effects because of their poor stability in vivo, short cycle time, and no targeting effect, which limit their role in tumor therapy [3]. Cisplatin (DDP) is a widely used chemotherapy drug for various types of cancer, including cervical cancer [3]. It is well known that metal–organic frameworks (MOFs) have many advantages, such as stable crystal structure, high drug-loading efficiency, good dispersion, and adjustable size, so they are widely used in the field of tumor drug delivery [4]. Among them, zeolitic imidazolate framework-8 (ZIF-8) has many advantages, such as specific degradation and good biocompatibility, etc. It has been widely used as a carrier of chemotherapeutic drugs, in order to improve the effect of chemotherapy on tumors [5]. Ma et al. synthesized a novel nano-composite (ZP-PDA-DOX) with the ability of tumor targeting [6]. ZIF-8 is degraded efficiently to release chemotherapeutic drugs when the nanoparticles are in a weak acidic tumor microenvironment, so as to eradicate tumor cells.

Photothermal therapy (PTT) is a new method of tumor therapy, which converts light energy into hyperthermia by photothermal converting agent for tumor ablation [7]. PTT was combined with chemotherapeutic drugs, and the drugs were loaded on different nano-carriers to realize on-demand drug release by photothermal effect. At the same time, the heat produced by the photothermic agent also increased the uptake of the drug by tumor cells, significantly improving the anti-cancer efficacy, and has a good targeting. A variety of photothermal materials with excellent PTT effect have been widely studied, including inorganic nanomaterials and polymer nanoparticles [8,9]. Most photothermic agents have shown great promise in the treatment of cancer; however, some properties, such as poor photostability, low efficiency of photothermal conversion, and short cycle time, hinder its further application in cancer therapy. Because of its excellent biocompatibility and biodegradability, polydopamine (PDA) nanomaterials have been regarded as excellent materials for biomedical applications [10]. Currently, PDA nanomaterials are widely used as PTT carriers due to their unique strong light absorption properties and high photothermal conversion efficiency [11]. Additionally, PTT can stimulate the immune system to release tumor antigens into the tumor microenvironment, activating T cells to achieve antitumor therapy [12].

Tumor tissues use immunosuppressive cells to form an immunosuppressive tumor microenvironment, which mediates immune tolerance and influences the therapeutic effect [13]. Therefore, the use of a nano-platform to suppress the tumor microenvironment can improve the efficacy of immunosuppression of cancer. Non-steroidal anti-inflammatory drugs (NSAIDs) have been widely used in clinic as antipyretic, analgesic, anti-inflammatory and anti-rheumatic drugs, and play an important role in the prevention of non-steroidal anti-inflammatory drug cerebrovascular disease [14]. However, a large number of epidemiological experiments and clinical studies in recent years have shown that long-term non-steroidal anti-inflammatory drugs can reduce the incidence of cancer. Research indicates that anti-inflammatory drugs can disrupt the tumor microenvironment by inhibiting cellular biological processes and enhancing chemotherapy sensitivity [15]. It has been shown that blocking the inflammatory signal pathway and modulating the immune response is an excellent choice for the treatment of cervical cancer [16].

In this paper, the polydopamine was wrapped around ZIF-8 to form a kind of near-infrared absorption nanoparticles. To achieve combination therapy, we used DDP as a chemotherapeutic agent and Aspirin-DL-Lysine (ADL) as an anti-inflammatory agent in combination with polydopamine-based nanoparticle photothermal therapy (Figure 1). This study provides the concept that a combination of photothermal therapy with chemotherapy and anti-inflammatory therapy will be achieved by ablation of the local tumor, robust strategies for the suppression of distant tumors with enhanced antitumor therapy outcomes.

## 2. Results and Discussion

### 2.1. Synthesis and Characterization of DDP@ZIF-8@PDA and DDP-ADL@ZIF-8@PDA

DDP@ZIF-8@PDA and DDP-ADL@ZIF-8@PDA were prepared via assembling ZIF-8, DDP, and ADL as shown in Figure 1. The dynamic light scattering (DLS) and transmission electron microscopy (TEM) of DDP@ZIF-8@PDA and DDP-ADL@ZIF-8@PDA nanoparticles were recorded and the results are shown in Figure 2A,B. The hydrodynamic diameters of DDP@ZIF-8@PDA and DDP-ADL@ZIF-8@PDA nanoparticles are about 500 nm and 300 nm, respectively. Compared to DDP@ZIF-8@PDA, the particle size of DDP-ADL@ZIF-8@PDA decreased. This reduction may result from interactions between the two drugs when loaded onto the same nanoparticle, leading to a more compact nanoparticle structure. Smaller nanoparticle size enhances stability and increases specific surface area, promoting drug dissolution. Consequently, DDP-ADL@ZIF-8@PDA loaded with dual nanodrugs may exhibit higher bioavailability. Compared to previous studies, the particle size difference is not significant, with particles uniformly distributed around 300 nm [17,18]. However, whereas the nanoparticles in earlier research carried a negative charge, those in this study are positively charged. Positively charged nanoparticles can more readily form electrostatic interactions with negatively charged cell membranes, thereby enhancing drug delivery efficiency [19]. We then examined the zeta potentials of DDP@ZIF-8@PDA and DDP-adl@ZIF-8@PDA (Figure 2C). The Zeta potentials of DDP@ZIF-8@PDA and DDP-ADL@ZIF-8@PDA were 4.1 ± 1.03 mV and 14.1 ± 1.72 mV, respectively. Qualitative analysis of the modification process of nanomedicines via FT-IR reflects interactions between different materials [17,20]. As shown in Figure 2D, ZIF-8, PDA, DDP@ZIF-8@PDA, and DDP-ADL@ZIF-8@PDA all exhibit distinct characteristic peaks in the range of 3600 cm^−1^ to 3100 cm^−1^, likely originating from the stretching vibrations of hydroxyl (OH) groups in the aqueous system. Concurrently, the characteristic peaks of PDA, DDP@ZIF-8@PDA, and DDP-ADL@ZIF-8@PDA between 2980 cm^−1^ and 2886 cm^−1^ correspond to C-H stretching vibrations, confirming the presence of methyl groups in all three materials. Furthermore, the characteristic peaks of ZIF-8, DDP@ZIF-8@PDA, and DDP-ADL@ZIF-8@PDA between 1700 cm^−1^ and 1760 cm^−1^ correspond to C=O stretching vibrations. These results indicate that the surfaces of DDP@ZIF-8@PDA and DDP-ADL@ZIF-8@PDA have been successfully modified with ZIF-8 and PDA, further confirming the successful preparation of the nanodrug.

### 2.2. Photothermal Effect of DDP-ADL@ZIF-8@PDA

As shown in Figure 3A, compared to the water sample temperature (<3 °C), the nanoparticles (200 μg/mL) reached a temperature of 54.1 °C within 5 min of laser irradiation. At the same time, the temperature increment of DDP-ADL@ZIF-8@PDA was dependent on its concentration (Figure 3A) and the power density of the NIR laser (Figure 3B), suggesting that temperature could be precisely controlled to induce mild hyperthermia. At the same time, no significant changes were observed after four cycles of exposure of DDP-ADL@ZIF-8@PDA to the NIR (Figure 3C), clearly demonstrating the excellent light stability of our prepared nanoparticles. In addition, we further measured the photothermal conversion efficiency (η) of DDP-ADL@ZIF-8@PDA nanoparticles by methods previously reported. The results showed that η of DDP-ADL@ZIF-8@PDA was 51.5%, which could be used as an ideal photothermal transforming agent for tumor PTT.

### 2.3. In Vitro Dual-Stimulation Responsive Release of DDP-ADL@ZIF-8@PDA

The encapsulation rate and drug-loading efficiency test results are shown in Appendix A. The encapsulation rate of DDP in DDP@ZIF-8@PDA and DDP-ADL@ZIF-8@PDA were 96.13 ± 2.33% and 91.09 ± 1.43%, respectively, while the drug-loading efficiencies were 11.33 ± 0.28% and 10.07 ± 0.28%, respectively. In formulation DDP-ADL@ZIF-8@PDA, ADL exhibited encapsulation and drug-loading efficiencies of 92.43 ± 0.85% and 10.21 ± 0.38%, respectively. Compared to previous PDA drug carrier studies [17], the encapsulation efficiency of the DDP-ADL@ZIF-8@PDA prepared in this study was slightly lower (98.9% vs. 91.09–92.43%), but the drug-loading capacity was relatively higher (5.7% vs. 10.07–10.21%). This discrepancy may be attributed to differences in the amount of drug loaded or variations in the preparation methods employed. It has been shown that the drug-loaded drug can be released from the carrier under the action of acid stimulation and mild high temperature. Therefore, we further evaluated the release behavior of DDP-ADL@ZIF-8@PDA nanoparticles under different conditions. As shown in Figure 3D, the amount of DDP released at pH 7.4 after 24 h was only about 9.6%, confirming that the prepared DDP-ADL@ZIF-8@PDA nanoparticles were highly stable in the blood circulation as well as in the normal physiological environment. At pH 5.0, 27.8% of DDP was released, which was due to ZIF-8 being sensitive to a weak acid environment and having a weak acid-responsive drug release function. In addition, the release behavior of DDP under near-infrared irradiation was also evaluated (Figure 3D). NIR thermal images (Figure 3E) show the temperature changes of different concentrations of DDP-ADL@ZIF-8@PDA nanoparticles after 10 min of laser irradiation at 808 nm at 2.0 W cm^−2^. Importantly, DDP release showed a unique NIR response pattern, with the released DDP being significantly accelerated by laser irradiation. The local high temperature induced by near-infrared laser can lead to the separation of PDA from DDP, which leads to the release of near-infrared responsive drugs. The tumor microenvironment has the characteristics of low pH value, and mild high temperature can be controlled accurately by adjusting the laser power density and irradiation time. Therefore, pH/NIR dual-stimulation-responsive DDP-ADL@ZIF-8@PDA nanoparticles may become an excellent antitumor drug.

### 2.4. In Vitro Combination of Chemo-PTT

First, the cytotoxicity of ZIF-8 on Hela cells was detected by CCK-8. As shown, even at high concentrations of 80 μg/mL, ZIF-8 treatment for 24 h showed no significant cytotoxicity in both cell lines. These results indicate that the ZIF-8 nanocarriers have low cytotoxicity and are suitable for biomedical applications (Figure 4A). Subsequently, the efficacy of DDP@ZIF-8@PDA nanoparticles in combination with PTT was studied. First, we evaluate the therapeutic effect of DDP@ZIF-8 on cervical cancer cells. Even at a high dose of 10 μg/mL, the inhibitory effect of the free DDP-treated group was not satisfactory (Figure 4A). In contrast, cells treated with DDP@ZIF-8 exhibited a DDP concentration-dependent reduction in cytotoxicity. Then, cells treated with PBS with or without laser irradiation maintained relative viability up to 94%, whereas cells treated with DDP@ZIF-8@PDA with laser irradiation had significantly reduced viability to 84.1%. These data indicate that the prepared nano-platform can be used as an effective drug delivery system for optimizing chemotherapy. More importantly, at relatively low concentrations of DDP (5 μg/mL), only less than 20% of cervical cancer cells survived after being treated with DDP@ZIF-8@PDA and NIR. The results show that the integrated DDP@ZIF-8@PDA nano-platform has significant antitumor activity. The combined index (CI) of chemotherapy and PTT was 0.17, which confirmed the synergistic effect. To visually demonstrate the antitumor efficacy of synergistic therapy, colony formation and live/dead cell staining assay was performed (Figure 4B,C). The results showed that after DDP@ZIF-8@PDA and NIR treatment, the cervical cancer cells showed strong red fluorescence signal, which was significantly different from other treatment groups. This result is consistent with that of CCK-8 and colony formation. Finally, cell Flow cytometry experiments were used to investigate the incidence of apoptosis induced by DDP@ZIF-8@PDA (200 μg/mL) and 808 nm laser irradiation (2.0 W cm^−2^, 10 min). As shown in Figure 4D, the apoptosis rate of cervical cancer cells treated with DDP@ZIF-8@PDA plus 808 nm laser irradiation was 98.58%, whereas the apoptosis rate of cervical cancer cells treated with free DDP or 808 nm laser alone was much lower than this. All these results strongly demonstrate that integrated DDP@ZIF-8@PDA has great potential for synergistic chemotherapy–PTT in cervical cancer.

### 2.5. In Vivo Combination of Chemo–Anti-Inflammatory–PTT

A bilateral Hela tumor model in mice was established to take advantage of the effect of combined therapy to ablate the primary tumor and suppress the growth of distant tumors (Figure 5A). In this study, Aspirin-DL-Lysine (ADL) was selected as an anti-inflammatory drug to prepare DDP-ADL@ZIF-8@PDA nanoparticles for combined therapy. As shown in Figure 5B,C, the tumor surface temperature of mice injected with DDP@ZIF-8@PDA or DDP-ADL@ZIF-8@PDA rapidly increased to ~45 °C within 1 min under 808 nm laser irradiation (2.0 W cm^−2^), and the temperature can rise to more than 50 °C in five minutes. The heat is high enough to kill the tumor cells. As shown in Figure 3D, the primary tumor volume of mice treated with NIR alone grew rapidly, exhibiting negligible antitumor effects. The administration of photothermal therapy and combination therapy based on DDP@ZIF-8@PDA resulted in significant suppression of the primary tumor (Figure 5D,E).

To our surprise, the PTT plus chemotherapy group showed only moderate distal tumor suppression, suggesting that PPT plus chemotherapy does not effectively suppress distal tumors. However, the combination of PPT with chemotherapy and anti-inflammatory therapy significantly inhibited the growth of distant tumors (Figure 5F,G). The tumor growth inhibition rate (TGI) of the combined treatment group (Group 7) was 60.3%, which was significantly higher than that of the other groups. All these results demonstrate that PTT plus anti-inflammatory therapy with DDP-ADL@ZIF-8@PDA nanoparticles has a higher therapeutic effect than PPT plus chemotherapy. No significant change in the weight of mice was observed during treatment, indicating that the adverse effects on mouse growth were negligible (Figure 5H).

### 2.6. The Enhanced Antitumor Mechanism of Chemo–Anti-Inflammatory–PTT

As shown in Figure 6A,B, after 12 and 24 h of treatment, *TNF-α* and *IL-6* levels were significantly increased in the PTT plus chemotherapy group. At the same time, the expressions of *TNF-α* and *IL-6* were down-regulated in PTT combined with chemotherapy and anti-inflammatory treatment groups because the inflammatory reaction was inhibited after ADL treatment. The levels of *TNF-α* and *IL-6* in all treatment groups recovered after 72 h, and there was no significant difference between the two groups. These results suggest that ADL in combination with PTT and chemotherapy can effectively inhibit the inflammatory reaction.

Interferon-γ (IFN-γ) is a cytokine that typically has biological activity in cellular suppression/cytotoxicity and antitumor mechanisms during cell-mediated adaptive immune responses [21]. In the antitumor mechanism, IFN-γ can induce apoptosis of regulatory T cells, stimulate the activity of M1 pro-inflammatory macrophages, and so on, and prevent tumor progression [22]. As shown in Figure 6C, *IFN-γ* levels were higher in the DDP-ADL@ZIF-8@PDA + NIR group than in the other treatment groups. *IFN-γ* level in NIR group was higher than that in non-NIR group, indicating that PTT can activate immunity. PTT combined with anti-inflammatory drugs had a stronger effect on immune activation. Forkhead transcription factor 3 (Foxp3), programmed death factor 1 (PD-1), and interleukin-10 (IL-10) are important immune negative regulatory molecules, and their high expressions are related to the suppression of cellular immunity [23]. As shown in Figure 6C, the expression of *IL-10* was increased in the DDP@ZIF-8@ PDA + NIR group due to immunosuppression, but the level of *IL-10* was decreased after DDP-ADL@ZIF-8@PDA + NIR combination treatment. Furthermore, *Foxp3* and *PD1* levels were lower in the DDP@ZIF-8@PDA + NIR group than in the DDP@ZIF-8@PDA or NIR group, respectively, whereas the expression of *Foxp3* and *PD1* was down-regulated in the DDP-ADL@ZIF-8@PDA + NIR group. The decrease in *IL-10*, *Foxp3* and *PD1* in the ADL@ZIF-8@ PDA + NIR group demonstrated that the immunosuppression was attenuated.

T lymphocyte (CD3+) is a major component of tumor immunity, which mediates the cellular immune process of the human body [24]. It mainly regulates the immune response process initiated by protein antigens and eliminates the corresponding antigens on the cell membrane or microorganisms in the cytoplasm. In the course of immune process, different T lymphocytes have different functions, mainly Helper T lymphocyte (Th, CD4+) and Cytotoxic T lymphocyte (TC, CD8+). The former has the ability to assist humoral and cellular immunity, while the latter has the function of killing target cells [24]. Dendritic cells (DC) are the most potent antigen-presenting cells that activate specific T lymphocytes, among which CD11c is the more specific surface antigen of mature DC cells, CD80 and CD86, as important costimulatory molecules to activate T lymphocytes, which also play an important role in DC cell maturation [25]. As shown in Figure 7C,D, the PTT combined with chemotherapy and anti-inflammatory therapy group induced a significantly increased number of CD8+ T cells compared to other groups, indicating that the combination therapy enhances the immune response. Furthermore, compared to other groups, the PTT combined with DDP and ADL group exhibited significantly up-regulated CD11c expression (Figure 7A), and the expression levels of CD80 and CD86 also increased after combination therapy (Figure 7B). These results suggest that PTT combined with chemotherapy and anti-inflammatory therapy can enhance the activation and maturation of the dendritic cell, activate the immune function, and improve the immune response.

As precursor cells of macrophages, bone marrow-derived monocytes (BMDMs), namely F4/80+ CD11B+ cells and their related products, play an important role in the growth and metastasis of tumor cells [26]. LY6G is an antigenic marker of mouse neutrophil [27]. Studies have shown that it plays an important role in neutrophil infiltration, recruitment, and migration [28]. We further investigated macrophage infiltration and neutrophil reduction in distant tumors in mice to confirm the exertion of tumor immunosuppression after combination therapy (Figure 7E,F). PTT in combination with chemotherapy increased the proportion of macrophages and neutrophil in distant tumors, while PTT in combination with chemotherapy and anti-inflammatory therapy decreased the infiltration of corresponding phenotype bone marrow cells. This demonstrates that anti-inflammatory benefits reduce tumor-associated immunosuppression.

In conclusion, these results demonstrate that PTT combined with chemotherapy and anti-inflammatory therapy can not only inhibit the expression of inflammatory factors, but also enhance the activation and maturation of DC. Treatment with increased anti-inflammatory drugs can reverse the immunosuppressive tumor microenvironment, promote tumor T cell infiltration, and enhance the immune response, thus enhancing the efficacy of PTT.

### 2.7. Biosafety Assessment

To further confirm the synergistic effect and potential toxicity of DDP-ADL@ZIF-8@PDA +NIR in vivo, we performed histological analysis in contralateral tumor tissue. As shown in Figure 8, fewer purple areas from H&E results were observed in the DDP-ADL@ZIF-8@PDA + NIR group, and larger areas of apoptosis were analyzed by TUNEL, indicating a satisfactory synergistic therapeutic effect. At the same time, HE staining was used for pathological analysis of the main organs. As shown in Appendix A, no tissue damage or inflammation occurred in the major organs, including the heart, liver, spleen, lungs, and kidneys, during the 14-day treatment period, which shows that this kind of integrated nanosystem has biocompatibility. All these systematic studies confirmed the prospect of DDP-ADL@ZIF-8@PDA as a synergistic antitumor therapy in vivo.

## 3. Materials and Methods

### 3.1. Materials

Zinc nitrate hexahydrate [Zn(NO_3_)_2_·6H_2_O] was purchased from national medicine, 2-methimidazole (2-MIM) from Sigma-Aldrich (Burlington, MA, USA), quercetin from Maclin, China, and phosphate buffer from HyClone, Logan, UT, USA. DDP was obtained from MedChemExpress (Shanghai, China, Cat. No. HY-17394), and ADL was obtained from MedChemExpress (China, Cat. No. HY-14654B).

### 3.2. Synthesis of DDP@ZIF-8@PDA and DDP-ADL@ZIF-8@PDA Nanoparticles

The ZIF-8 nanoparticles were prepared according to previous reports. Zn(NO_3_)_2_·6H_2_O (150 mg) was dissolved in H_2_O (5 mL), designated as solution A, and 2-MIM (330 mg) with or without DDP (100 mg) and ADL (100 mg) in MeOH (10 mL), designated as solution B. Then, both solutions A and B are left under magnetic stirring until the powder therein is completely dissolved. After that, solution A was rapidly transferred to solution B, and the reaction was maintained for 15 min. Finally, the white solid precipitate from the reaction was collected after centrifugation (10,000 rpm, 15 min), washed several times with MeOH, and then dried under 60 °C vacuum overnight to completely remove the residual MeOH in the white solid.

A total of 1.0 g of DDP@ZIF-8 and DDP-ADL@ZIF-8 was added into 100 mL Tris solution (pH = 8.5). A total of 0.16 M dopamine was subsequently added into the mixed solution. Then, the mixed solution was stirred for 6 h at room temperature, then centrifuged and washed several times with deionized water. In this process, the color of the solution changed from clear to black. The solution was then dried under 60 °C vacuum to obtain DDP@ZIF-8@PDA and DDP-ADL@ZIF-8@PDA powders.

### 3.3. Characterization

Dissolve an appropriate amount of nanomaterials in methanol, dilute the solution, and deposit 2 μL of the diluted solution onto a copper grid. Observe the morphology using a transmission electron microscope (TEM). A nanoparticle size potential analyzer was used to measure the surface charge of nanomaterials in water solution. In order to obtain the chemical bond and functional group information of nanomaterials, the appropriate amount of nanomaterials powder and KBr powder was fully ground by Fourier-transform infrared spectroscopy (FTIR) scan to determine the characteristic functional group information of the above three samples.

### 3.4. Measurement of Photothermal Performance

DDP-ADL@ZIF-8@PDA dispersions (200 μL) with different concentrations (50–200 μg/mL) were exposed to 808 nm near-infrared laser for 5 min at a power density of 1.0 W cm^−2^. In addition, DDP-ADL@ZIF-8@PDA dispersions at a concentration of 100 μg/mL were exposed to lasers with different power densities (0.5, 0.8, 1.0, and 1.2 W cm^−2^) for 5 min. Infrared Thermal Imaging Camera (Fluke, Everett, WA, USA) was used to monitor the temperature changes of different dispersions. To verify the photothermal stability of DDP-ADL@ZIF-8@PDA nanoparticles, 200 μL of dispersion (100 μg/mL) was irradiated with an 808 nm laser (1.0 W cm^−2^) for 5 min. The open/close step repeats the four loops. Then, the photothermal conversion efficiency (η) of DDP-ADL@ZIF-8@PDA nanoparticles was calculated according to the literature.

### 3.5. In Vitro Drug Release

Add 1 mL of DDP-ADL@ZIF-8@PDA solution (1.0 mg mL^−1^) to a dialysis bag (10 kDa), seal it, and place it in PBS solution (pH = 5.0 or 7.4). At different time points, remove 0.5 mL of the released solution and measure the absorbance using a UV-visible spectrophotometer (490 nm). To evaluate near-infrared (NIR)-triggered release of DDP and ADL, absorbance was measured using the above method after 10 min, 6 h, and 12 h exposure to NIR laser (1.0 W cm^−2^). Encapsulation efficiency = (Total drug loaded − Free drug)/Total drug loaded × 100%. Drug-loading efficiency = (Total drug loaded − Free drug)/Total nanoparticles weight × 100%.

### 3.6. In Vitro Biodegradation Evaluation of DDP-ADL@ZIF-8@PDA

For in vitro degradation behavior, 1.0 mg of DDP-ADL@ZIF-8@PDA was dispersed into simulated body fluid (SBF) solution (30 mL). The 1 mL DDP-ADL@ZIF-8@PDA dispersions were extracted and analyzed by transmission electron microscopy (TEM) and inductively coupled plasma emission spectroscopy (ICP-AES) on days 1, 3, 5, and 7, respectively.

### 3.7. In Vitro PA Imaging

Different concentrations (0–10 mM) of DDP-ADL@ZIF-8@PDA dispersions were filled into a single well on agar plates, and in vitro photoacoustic (PA) imaging performance was then assessed by the Vevo LAZR PA system (VisualSonics Inc., Toronto, ON, Canada) at a wavelength of 808 nm.

### 3.8. In Vitro Cytotoxicity Assay of Nanoparticles

Hela (BNCC, BNCC342189) and C33A (BNCC, BNCC354329) cells were cultured in DMEM medium. Cervical cancer cells (1 × 10^4^) were seeded into a 96-well plate. After 6 h of incubation, 100 μL of medium containing DDP-ADL@ZIF-8@PDA (0–100 μg/mL) was added. Cell viability was assessed using the CCK-8 assay kit (Beyotime Biotechnology, Shanghai, China) after 24 h.

### 3.9. In Vitro Cellular Uptake of DDP-ADL@ZIF-8@PDA Nanoparticles

Seed Hela cells (1 × 10^5^) into culture dishes. After cells adhere, remove the medium and replace it with either DDP-free medium or medium containing DDP-ADL@ZIF-8@PDA. Cells co-cultured with DDP-ADL@ZIF-8@PDA nanoparticles were irradiated with an 808 nm laser for 5 min. After 2 h, cells were rinsed with phosphate-buffered saline (PBS), then stained with 4′,6-diamidino-2-phenylindole (DAPI) for 30 min and observed using confocal microscopy. In addition, Flow cytometry was used to measure the intracellular DDP fluorescence.

### 3.10. In Vitro Chemo–Anti-Inflammatory–PTT Synergistic Effect

Hela cells were seeded in 96-well plates (1 × 10^4^ wells) and cultured for 6 h. After that, 100 μL fresh media containing different concentrations of free DDP, ZIF-8, DDP@ZIF-8, DDP-ADL@ZIF-8, or DDP-ADL@ZIF-8@PDA were added to replace the previous media. After further incubation for 24 h, cell viability was determined by CCK-8 assay. To evaluate synergistic effects, Hela cells were cultured with 100 μL DDP-ADL@ZIF-8@PDA, and some cells were irradiated with laser (808 nm, 1 W cm^−2^) for 5 min. Cell viability was measured by CCK-8 after 24 h of culture. At the same time, a combined index (CI) analysis evaluated the synergistic effects of chemotherapy, anti-inflammatory therapy, and photothermal therapy.

### 3.11. Live/Dead Assay

Cells were stained with calcein-AM (5 μg/mL) and PI (10 μg/mL) for 20 min. The staining results were observed under a fluorescence microscope to further evaluate the antitumor efficacy of the nanomedicine.

### 3.12. Animal and Tumor Models

Five-week-old female BALB/C mice were purchased from Shanghai SLAK Laboratory Animal Co., Ltd., Shanghai, China, Hela cells (1 × 10^6^) were injected subcutaneously into the right hindlimb of the mice. When the tumor volume reached approximately 50 mm^3^, the same cells were injected into the left hindlimb. All mice were randomly divided into seven groups: (1) ZIF-8, (2) NIR, (3) free DDP, (4) DDP@ZIF-8@PDA, (5) DDP @ZIF-8@PDA + NIR, (6) DDP-ADL@ZIF-8@PDA, and (7) DDP-ADL@ZIF-8@PDA + NIR. Subsequently, 100 μL of the nanodrug was injected into the primary tumor. In groups (1), (2), (3), (4), (5), and (7), the drug concentrations were 0.4 mg/mL. In addition, mice in (2), (5), and (7) groups were irradiated with a laser for 5 min after injection. After a 14-day treatment period, organ and tumor samples were harvested from the mice for hematoxylin and eosin (H&E) staining. This was conducted to evaluate the efficacy and biological safety of the combination therapy.

### 3.13. Detection of Cytokines

In order to investigate the levels of inflammatory factors in vivo, serum samples of mice were extracted at 12, 24, and 72 h after different treatments. An Elisa kit was used to determine the expression levels of tumor necrosis factor α (TNF-α) and IL-6. Quantitative PCR was used to detect the expression of cytokines IL-10, FOXP3, PD1, and IFN-γ.

### 3.14. Detection of Antibody Expression in Distal Tumors

Freshly isolated distal tumor tissue was minced and digested at 37 °C for 2 h. Single-cell suspension was obtained by filtration, centrifugation, and washing. For metastatic nodules, they were minced and digested at 37 °C for 10 min. After filtration, the immune cells were removed by gradient centrifugation with Percoll (Cytiva). About 100 μL of a monoclonal antibody dye mixture with multiple fluorescences were added to each sample. The monoclonal antibodies were used as follows: anti-mouse CD11c APC (BioLegend (San Diego, CA, USA), cat#: 117310), anti-mouse CD80 PE (BioLegend, cat#: 104707), anti-mouse CD86 FITC (Thermo (Waltham, MA, USA), cat#: 11-0862-82), anti-mouse MHC II eF450 (Thermo, cat#: 48-5321-82), anti-mouse CD45 BV510 (BioLegend, cat#: 103138), Fixable Viability Dye eFluor780 (Thermo, cat#: 65-0865-14), anti-mouse CD3 PE-Cy7 (BioLegend, cat#: 100320), anti-mouse CD8 PerCP-Cy5.5 (Thermo, cat#: 45-0081-82), anti-mouse CD11b FITC (BioLegend, cat#: 101205), anti-mouse Ly6G PE (Thermo, cat#: 12-9668-82), anti-mouse F480 APC (BioLegend, cat#: 123116), and anti-mouse CD45 BV421 (BioLegend, cat#: 103134). All samples were determined by BD LSRFortessa^TM^ X-20 (BD Biosciences, New Jersey, USA) cell analyzer.

### 3.15. Statistical Analysis

Data statistical analysis and graphing were performed using Origin 2024 and GraphPad Prism 9.5.0. Experimental results are expressed as mean ± standard deviation. * *p* < 0.05 is considered statistically significant.

## 4. Conclusions

We report DDP-ADL@ZIF-8@PDA nanomedicine for effective drug delivery, synergistic photothermal therapy, and anti-inflammatory attenuated immunotherapy to enhance treatment of human cervical cancer xenografts in mice. DDP-ADL@ZIF-8@PDA has a cavity structure, good drug-loading performance, and pH/NIR laser dual-reaction drug release behavior. The formed DDP-ADL@ZIF-8@PDA nanoparticles possess strong near-infrared absorption ability, and can effectively convert light energy into light–heat with a photothermal conversion efficiency of up to 51.5%. Induced mild hyperthermia can significantly increase cellular uptake of nanoparticles to enhance therapeutic efficacy. Importantly, DDP-ADL@ZIF-8@PDA + NIR was shown to inhibit the primary and distant tumors of cervical carcinoma in mice. PTT combined with chemotherapy and anti-inflammatory therapy can enhance the activation of dendritic cells, attenuate immunosuppression, promote the infiltration of immune cells, and increase the activity of antitumor CTL. In addition, the nanodrugs are designed to be biocompatible and less toxic. Therefore, this kind of nanodrug may provide a new idea for the treatment of cervical cancer.

## Figures and Tables

**Figure 1 pharmaceuticals-18-01657-f001:**
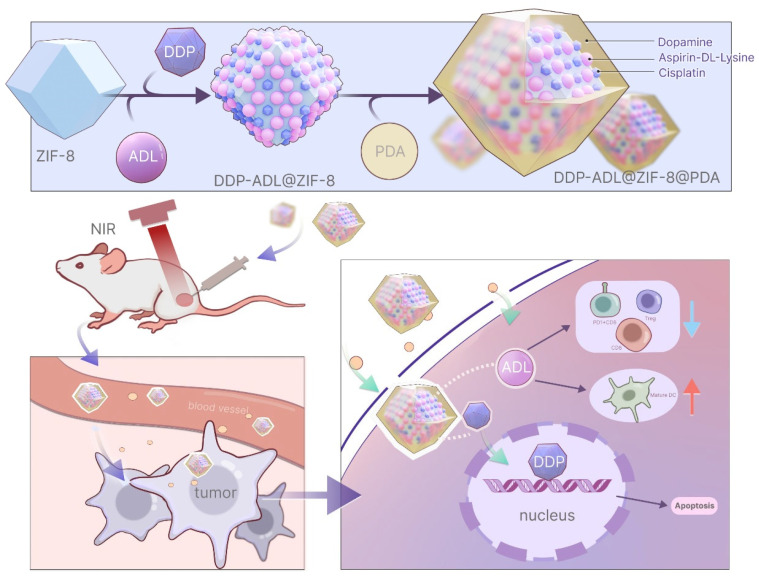
Synthesis of DDP-ADL@ZIF-8@PDA nanoparticles and schematic diagram of the mechanism of DDP-ADL@ZIF-8@PDA as a nanothermal platform for in vivo PTT combined with chemotherapy and anti-inflammatory therapy. The blue arrow indicates a downward adjustment, while the red arrow indicates an upward adjustment.

**Figure 2 pharmaceuticals-18-01657-f002:**
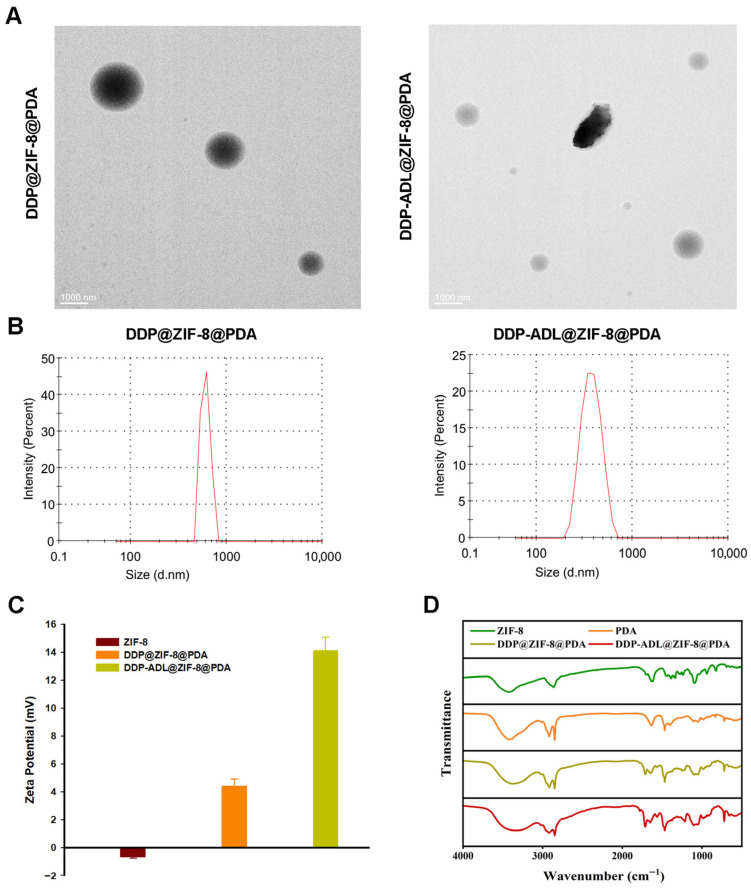
Characterization of materials. (**A**) TEM images of DDP@ZIF-8@PDA and DDP-ADL@ZIF-8@PDA nanoparticles. (**B**) The DLS of nanoparticles. (**C**) Zeta potentials of Zif-8, DDP@ZIF-8@PDA, and DDP-ADL@ZIF-8@PDA. (**D**) FTIR spectra of nanoparticles.

**Figure 3 pharmaceuticals-18-01657-f003:**
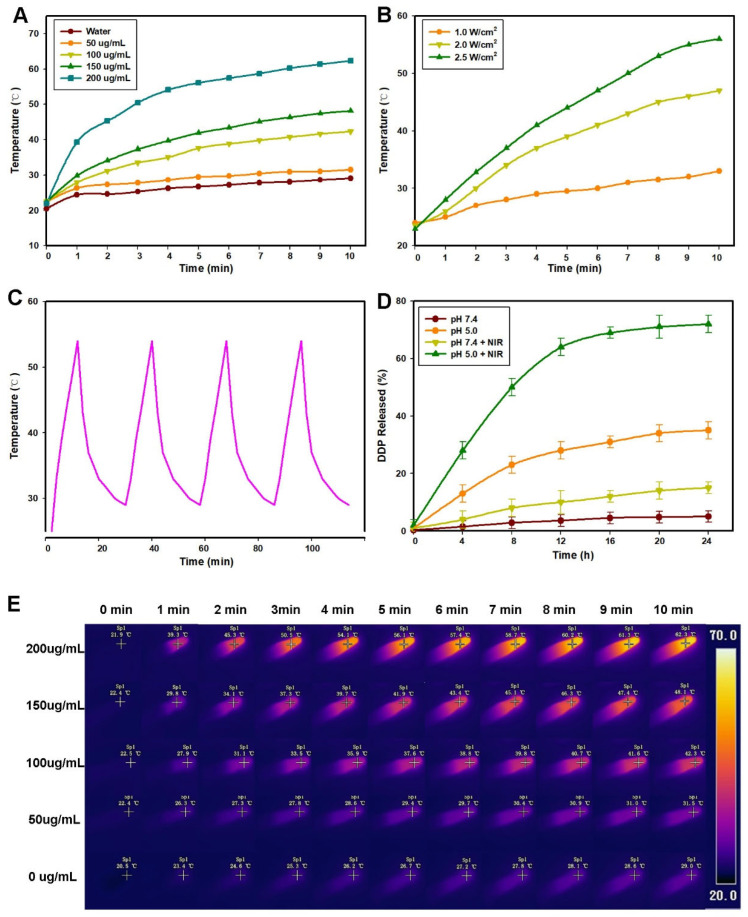
The photothermal curve and thermal imaging of materials. (**A**) Concentration-dependent photothermal curves of DDP-ADL@ZIF-8@PDA aqueous solution (808 nm laser, 1.0 W cm^−2^). (**B**) Laser power-dependent photothermal curves of DDP-ADL@ZIF-8@PDA aqueous solution (808 nm laser, 100 ug ML^−1^). (**C**) Temperature recordings of DDP-ADL@ZIF-8@PDA nanoparticle aqueous solution (200 ug mL^−1^) during four cycles of laser on/off at 1.0 W cm^−2^. (**D**) In vitro DDP release profiles from DDP-ADL@ZIF-8@PDA, with or without NIR at different pH values. (**E**) Thermal images of DDP-ADL@ZIF-8@PDA NPs solutions exposed to 808 nm laser.

**Figure 4 pharmaceuticals-18-01657-f004:**
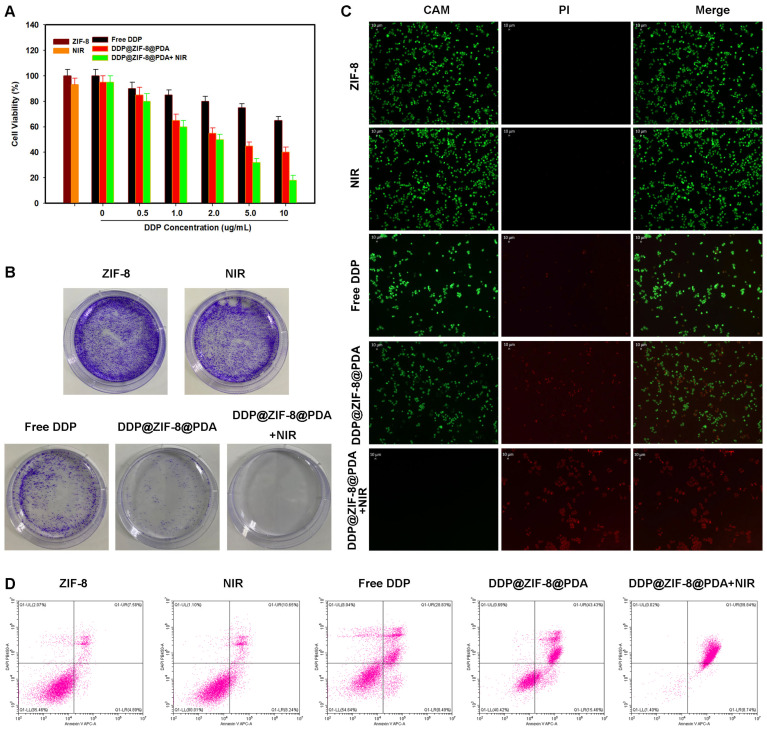
The impact on the survival of tumor cells of photothermal therapy combined with chemotherapy based on ZIF-8@PDA nanoparticles. (**A**) Relative viabilities of Hela cells after different treatments, including ZIF-8, laser irradiation only, free DDP only, and DDP@ZIF-8@PDA with or without laser irradiation. (**B**) Representative of colony formation in ZIF-8, laser irradiation only, free DDP only, and DDP@ZIF-8@PDA with or without laser irradiation. (**C**) Live/dead staining of Hela cells treated with ZIF-8, laser irradiation only, free DDP only, and DDP@ZIF-8@PDA with or without laser irradiation. (**D**) Hela cells treated with ZIF-8, laser irradiation only, free DDP only, and DDP@ZIF-8@PDA with or without laser irradiation were stained with Annexin-V/PI for Flow cytometric analysis.

**Figure 5 pharmaceuticals-18-01657-f005:**
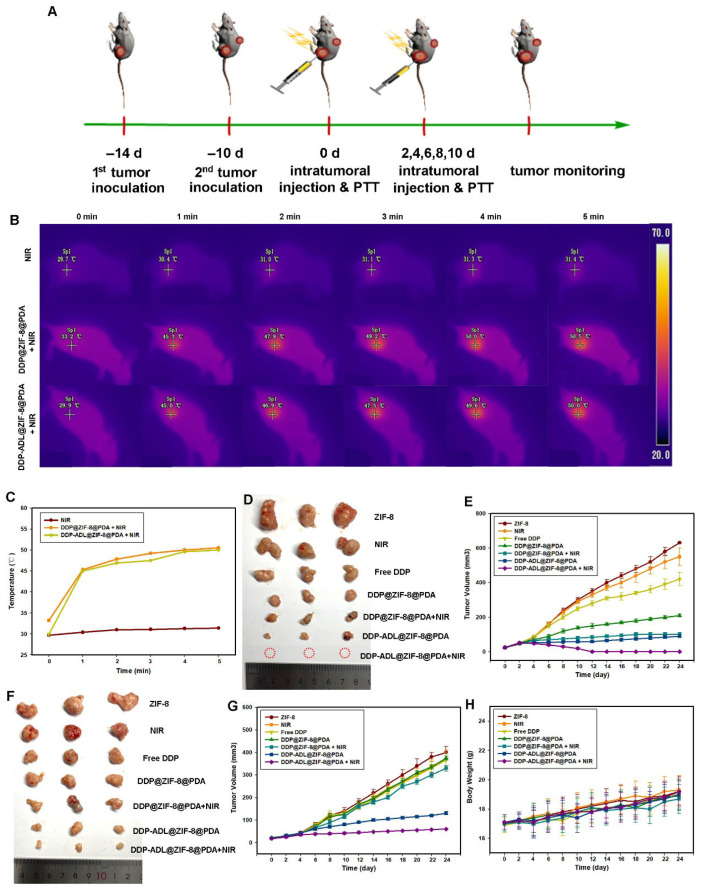
Antitumor effect of photothermal therapy combined with chemotherapy and anti-inflammatory therapy based on ZIF-8@PDA nanoparticles in a bilateral Hela tumor model. (**A**) Schematic diagram of the establishment of a bilateral Hela tumor-bearing mouse model of antitumor effect and immune activation. (**B**,**C**) Temperature of tumor tissue after 808 nm laser irradiation. (**D**,**E**) The tumor weight and photo of the dissected primary tumor. (**F**,**G**) The tumor weight and photo of the dissected distant tumor. (**H**) Body weight change curves in different groups.

**Figure 6 pharmaceuticals-18-01657-f006:**
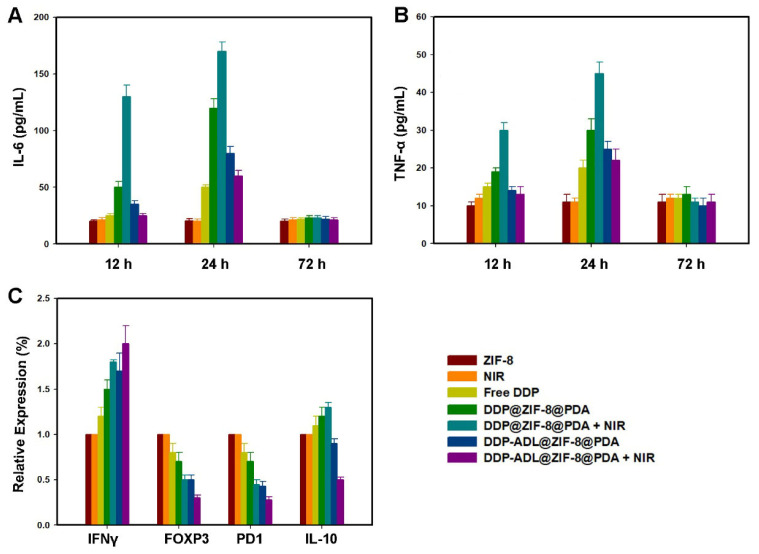
The inflammatory reaction induced by PTT and the anti-inflammatory effect of ADL. (**A**) *IL-6* levels in sera of mice after various treatments. (**B**) *TNF-α* levels in sera of mice after various treatments. (**C**) Cytokine levels in the distant tumor after various treatments.

**Figure 7 pharmaceuticals-18-01657-f007:**
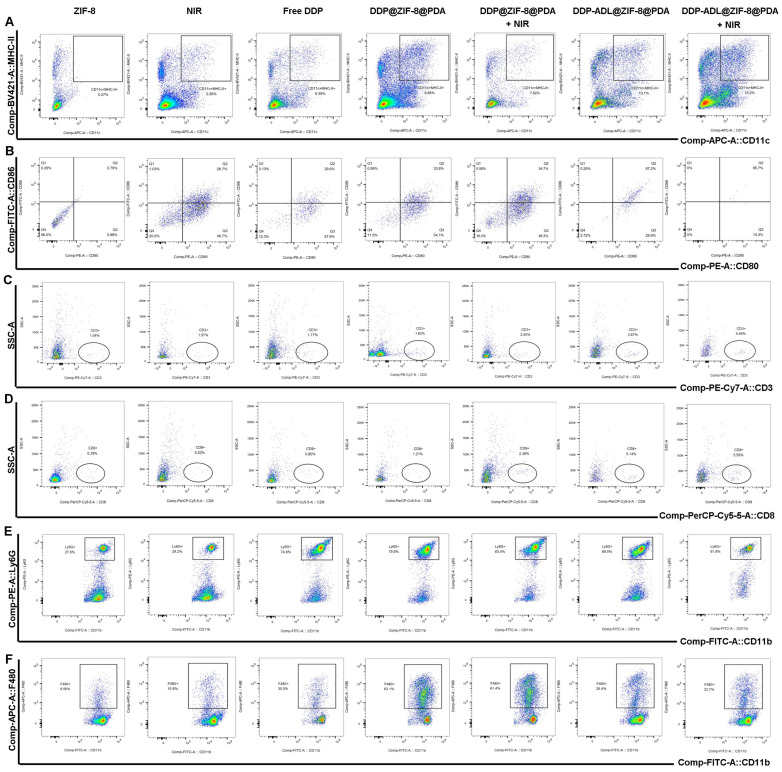
The inflammatory reaction induced by PTT and the anti-inflammatory effect of ADL. (**A**–**D**) Antigen presentation by the dendritic cells after various treatments. (**E**) Neutropenia infiltration in the distant tumor after various treatments. (**F**) Macrophage infiltration in the distant tumor after various treatments.

**Figure 8 pharmaceuticals-18-01657-f008:**
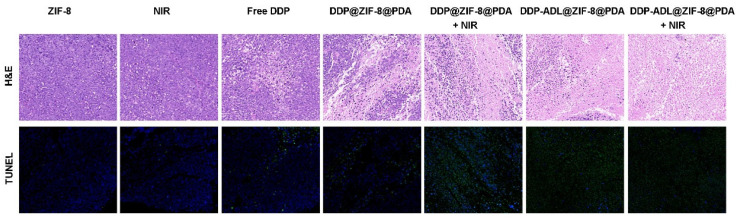
Representative histological analysis, including H & E and TUNEL staining of contralateral tumor sections after different treatments (200 magnification).

## Data Availability

Research data are not shared.

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
