# Peer review of "Photothermal Therapy Combined with Chemotherapy and Anti-Inflammation Therapy Weakens the Immunosuppression of Cervical Cancer"

_pharmaceuticals, 2025, doi:10.3390/ph18111657_

Round 1

Reviewer 1 Report

Comments and Suggestions for Authors

The manuscript "Photothermal Therapy Combined with Chemotherapy and Anti-Inflammation Therapy Weakens the Immunosuppression of Cervical Cancer" is devoted to the synthesis of nanomaterials based on the zeolitic imidazolate framework-8 (ZIF-8) loaded with the chemotherapeutic drug cisplatin (DDP) and the anti-inflammatory drug Aspirin-DL-Lysine (ADL) with a polydopamine (PDA) coating. A synergistic effect of the combination of photothermal therapy (PTT) with chemotherapy and anti-inflammatory therapy was showed. Of particular interest is the demonstration that PTT combined with DDP and ADL can reduce inflammation and the immunosuppressive tumor microenvironment and enhance the anti-tumor effect. However, a drawback of the manuscript is the insufficient characterization of the synthesized materials.

  1. The synthesis method for DDP@ZIF-8 and DDP-ADL@ZIF-8 does not indicate the amounts of DDP and ADL added to the reaction mixture. There is no information on the loading level of these particles with the drugs. It is also unknown what amount of the drug remains after PDA coating (in DDP@ZIF-8@PDA and DDP-ADL@ZIF-8@PDA).
  2. The size distribution of DDP@ZIF-8@PDA and DDP-ADL@ZIF-8@PDA based on TEM data is not provided. How do the particle sizes according to TEM data compare with those according to DLS?
  3. It would be interesting to compare the size and ζ-potential of the obtained particles with other known PDA-based materials using for drug delivery.
  4. There is no discussion of the IR spectra of the obtained materials. The spectra of ZIF-8 and the particles before PDA coating should also be included; these should be discussed.
  5. In Fig. 2D, IR spectra are shown instead of UV spectra.
  6. The work does not reference works after [16].
  7. Why was aspirin-DL-lysine chosen over aspirin (or another drug) as a nonsteroidal anti-inflammatory drug?

The manuscript may be recommended for publication in Pharmaceuticals, but only after the above-described issues have been resolved.

Author Response

Comments and Suggestions for Authors

The manuscript "Photothermal Therapy Combined with Chemotherapy and Anti-Inflammation Therapy Weakens the Immunosuppression of Cervical Cancer" is devoted to the synthesis of nanomaterials based on the zeolitic imidazolate framework-8 (ZIF-8) loaded with the chemotherapeutic drug cisplatin (DDP) and the anti-inflammatory drug Aspirin-DL-Lysine (ADL) with a polydopamine (PDA) coating. A synergistic effect of the combination of photothermal therapy (PTT) with chemotherapy and anti-inflammatory therapy was showed. Of particular interest is the demonstration that PTT combined with DDP and ADL can reduce inflammation and the immunosuppressive tumor microenvironment and enhance the anti-tumor effect. However, a drawback of the manuscript is the insufficient characterization of the synthesized materials.

  1. The synthesis method for DDP@ZIF-8 and DDP-ADL@ZIF-8 does not indicate the amounts of DDP and ADL added to the reaction mixture. There is no information on the loading level of these particles with the drugs. It is also unknown what amount of the drug remains after PDA coating (in DDP@ZIF-8@PDA and DDP-ADL@ZIF-8@PDA).

Reply: Thank you very much for your valuable comments. In the revised manuscript, we have incorporated your suggestion to include the exact amounts of DDP and ADL in the methodology section, specifically as “100 mg”. Additionally, we have supplemented the encapsulation efficiency and drug loading test results for DDP and ADL as per your recommendation to further clarify the drug loading levels and drug residue amounts. The specific additions are as follows:

The encapsulation efficiency and drug loading efficiency test results are shown in Figure S1A-B. The encapsulation rate of DDP in DDP@ZIF-8@PDA and DDP-ADL@ZIF-8@PDA were 96.13±2.33% and 91.09±1.43%, respectively, while the drug loading efficiencies were 11.33±0.28% and 10.07±0.28%, respectively. In formulation DDP-ADL@ZIF-8@PDA, ADL exhibited encapsulation and drug loading efficiencies of 92.43 ± 0.85% and 10.21 ± 0.38%, respectively.

Figure S1. Drug encapsulation efficiency and loading capacity testing. A. DDP and ADL encapsulation rate test; B. DDP and ADL loading efficiency testing.

  1. The size distribution of DDP@ZIF-8@PDA and DDP-ADL@ZIF-8@PDA based on TEM data is not provided. How do the particle sizes according to TEM data compare with those according to DLS?

Reply: Thank you very much for your question. The particle sizes measured by TEM data are consistent with those determined by DLS. The diameters of DDP@ZIF-8@PDA and DDP-ADL@ZIF-8@PDA are approximately 500 nm and 300 nm, respectively. In the original manuscript, we had already provided the DLS test data, which was originally Figure 2C (now Figure 2B in the revised version). However, due to an oversight on our part, the incorrect labeling of Figure 2B,C caused confusion, for which we sincerely apologize. In the revised manuscript, we have adjusted the position of Figure 2B,C.

  1. It would be interesting to compare the size and ζ-potential of the obtained particles with other known PDA-based materials using for drug delivery.

Reply: Thank you very much for your valuable suggestions. In the revised manuscript, we have incorporated your recommendation by adding a comparison with drug delivery materials based on previous PDA studies. The specific additions are as follows:

Compared to previous studies, the particle size difference is not significant, with particles uniformly distributed around 300 nm [17,18]. However, whereas the nanoparticles in ear-lier research carried a negative charge, those in this study are positively charged. Positive-ly charged nanoparticles can more readily form electrostatic interactions with negatively charged cell membranes, thereby enhancing drug delivery efficiency [19].

  1. There is no discussion of the IR spectra of the obtained materials. The spectra of ZIF-8 and the particles before PDA coating should also be included; these should be discussed.

Reply: Thank you very much for your suggestions. In the revised manuscript, we have re-tested and analyzed the infrared spectra of all materials, including ZIF-8 and PDA, based on your recommendations, and further discussed the test results. The specific additions are as follows:

Qualitative analysis of the modification process of nanomedicines via FT-IR reflects inter-actions between different materials [17, 20]. As shown in Figure 2D, ZIF-8, PDA, DDP@ZIF-8@PDA, and DDP-ADL@ZIF-8@PDA all exhibit distinct characteristic peaks in the range of 3600 cm−1 to 3100 cm−1, likely originating from the stretching vibrations of hydroxyl (OH) groups in the aqueous system. Concurrently, the characteristic peaks of PDA, DDP@ZIF-8@PDA, and DDP-ADL@ZIF-8@PDA between 2980 cm⁻¹ and 2886 cm⁻¹ correspond to C-H stretching vibrations, confirming the presence of methyl groups in all three materials. Furthermore, the characteristic peaks of ZIF-8, DDP@ZIF-8@PDA, and DDP-ADL@ZIF-8@PDA between 1700 cm−1 and 1760 cm−1 correspond to C=O stretching vibrations. These results indicate that the surfaces of DDP@ZIF-8@PDA and DDP-ADL@ZIF-8@PDA have been successfully modified with ZIF-8 and PDA, further confirming the successful preparation of the nanodrug.

(D) FTIR spectra of nanoparticles.

  1. In Fig. 2D, IR spectra are shown instead of UV spectra.

Reply: Thank you very much for pointing out the error. We have corrected it in the revised manuscript.

  1. The work does not reference works after [16].

Reply: Thank you very much for your comments. In the original manuscript, we have already cited references 17-24 in the “Results and Discussion” section.

  1. Why was aspirin-DL-lysine chosen over aspirin (or another drug) as a nonsteroidal anti-inflammatory drug?

Reply: Thank you very much for your question. Compared to aspirin or other drugs, ADL causes relatively less irritation to the gastrointestinal tract and dissolves easily. Therefore, we selected ADL as the drug model for our research.

The manuscript may be recommended for publication in Pharmaceuticals, but only after the above-described issues have been resolved.

Reviewer 2 Report

Comments and Suggestions for Authors

The authors describe the preparation of nanohybrids associating ZIF-8, polydopamine (PDA), the cisplatin (DDP) drug and the anti-inflammatory aspirin-DL-Lysine (ADL) drug and their use for combined photothermal and chemotherapy. The biological part of the manuscript is of interest but the characterization of the nanohybrids used for the study must be markedly improved. Moreover, along the whole manuscript, results are poorly discussed in the context of literature. The following comments should be considered :

  • clarify all the abbreviations used in the abstract and specifiy that DDP is cisplatin.
  • the novelty of the work should be highlighted at the end of the introduction.
  • paragraph 2: describe the preparation of DPP@ZIF-8 and of DPP-ADL@ZIF-8 hybrids. Indicate how much drug has been encapsulated.
  • figure 2 : TEM images are of modest quality. New TEM images on which a greater number of particles is present should be provided. The morphology of the hybrids should also be compared to pure ZIF-8. 
  • the authors should also discuss the decrease in diameter observed for DDP-ADL@ZIF-8@PDA compared to DDP@ZIF-8@PDA. These results must be verified. How many particles were counted ? A size distribution should be added for each sample.
  • FT-IR results are not described in the text.
  • UV-visible absorption spectra should be provided (at least for DDP@ZIF-8@PDA and DDP-ADL@ZIF-8@PDA). Why was an excitation wavelength of 808 nm used to evaluate the photothermal properties of the nanohybrids ?
  • what is the initial DDP charge in the nanoparticles ? the release kinetics of DDP as well as ADL should be studied and the results discussed in the context of literature. The advances made should be highlighted.

Author Response

Comments and Suggestions for Authors

The authors describe the preparation of nanohybrids associating ZIF-8, polydopamine (PDA), the cisplatin (DDP) drug and the anti-inflammatory aspirin-DL-Lysine (ADL) drug and their use for combined photothermal and chemotherapy. The biological part of the manuscript is of interest but the characterization of the nanohybrids used for the study must be markedly improved. Moreover, along the whole manuscript, results are poorly discussed in the context of literature. The following comments should be considered :

  1. clarify all the abbreviations used in the abstract and specifiy that DDP is cisplatin.

Reply: Thank you very much for your suggestion. In the revised manuscript, we have added the drug name represented by DDP to the abstract as per your recommendation.

  1. the novelty of the work should be highlighted at the end of the introduction.

Reply: Thank you very much for your suggestion. We have highlighted the novelty of this study at the end of the introduction. Specifically, “this study provides the concept that a combination of photothermal therapy with chemotherapy and anti-inflammatory therapy will be achieved by ablation of the local tumor, robust strategies for the suppression of distant tumors with enhanced anti-tumor therapy outcomes.”.

  1. paragraph 2: describe the preparation of DPP@ZIF-8 and of DPP-ADL@ZIF-8 hybrids. Indicate how much drug has been encapsulated.

Reply: Thank you very much for your suggestion. In the revised manuscript, we have added the drug dosage to Section 2.2. Specifically “100mg”.

  1. figure 2 : TEM images are of modest quality. New TEM images on which a greater number of particles is present should be provided. The morphology of the hybrids should also be compared to pure ZIF-8.

Reply: Thank you very much for your suggestions. In the revised manuscript, we have adjusted the clarity and enhanced the resolution of the TEM images to the greatest extent possible. When capturing TEM images, we exclusively used high-magnification TEM images to more clearly display the morphology of the particles. The current TEM images already show multiple nanoparticles, and we sincerely apologize for being unable to provide TEM images of additional particles. Furthermore, our objective in capturing TEM images was to provide a more intuitive and clear depiction of the morphology of the drug-loaded nanoparticles. Therefore, this study does not include comparative analysis with pure ZIF-8. Pure ZIF-8 is a commercially available product, and its morphology can be referenced in published research literature, such as: doi: 10.2147/IJN.S340764.

  1. the authors should also discuss the decrease in diameter observed for DDP-ADL@ZIF-8@PDA compared to DDP@ZIF-8@PDA. These results must be verified. How many particles were counted ? A size distribution should be added for each sample.

Reply: Thank you very much for your valuable comments. We have noted this phenomenon. The reason may be that when the two drugs are loaded into the same nanoparticle, interactions occur, leading to a more compact nanoparticle structure and consequently reducing the particle size. However, this hypothesis requires further verification. In the revised manuscript, we have added relevant discussion based on your suggestions. The specific additions are as follows:

Compared to DDP@ZIF-8@PDA, the particle size of DDP-ADL@ZIF-8@PDA decreased. This reduction may result from interactions between the two drugs when loaded onto the same nanoparticle, leading to a more compact nanoparticle structure. Smaller nanoparticle size enhances stability and increases specific surface area, promoting drug dissolution. Consequently, DDP-ADL@ZIF-8@PDA loaded with dual nanodrugs may exhibit higher bioavailability.

Additionally, we are unable to count the number of nanomedicine particles in solution. Therefore, we cannot answer your question regarding the count of particles. The particle size mentioned in the results is not an average derived from measuring multiple particles. It is based on dynamic light scattering (DLS) testing. The particle size distribution of the sample obtained via DLS testing is shown in Figure 2C (new Figure 2B). In the previous manuscript, confusion arose due to incorrect labeling of Figure 2B,C. This has been corrected in the revised manuscript. The current Figure 2B represents the particle size distribution.

  1. FT-IR results are not described in the text.

Thank you very much for your feedback. In the revised manuscript, we have incorporated your suggestion by adding an analysis of the FTIR results in Section 3.1. The specific content is as follows:

Qualitative analysis of the modification process of nanomedicines via FT-IR reflects inter-actions between different materials [17, 20]. As shown in Figure 2D, ZIF-8, PDA, DDP@ZIF-8@PDA, and DDP-ADL@ZIF-8@PDA all exhibit distinct characteristic peaks in the range of 3600 cm−1 to 3100 cm−1, likely originating from the stretching vibrations of hydroxyl (OH) groups in the aqueous system. Concurrently, the characteristic peaks of PDA, DDP@ZIF-8@PDA, and DDP-ADL@ZIF-8@PDA between 2980 cm⁻¹ and 2886 cm⁻¹ correspond to C-H stretching vibrations, confirming the presence of methyl groups in all three materials. Furthermore, the characteristic peaks of ZIF-8, DDP@ZIF-8@PDA, and DDP-ADL@ZIF-8@PDA between 1700 cm−1 and 1760 cm−1 correspond to C=O stretching vibrations. These results indicate that the surfaces of DDP@ZIF-8@PDA and DDP-ADL@ZIF-8@PDA have been successfully modified with ZIF-8 and PDA, further confirming the successful preparation of the nanodrug.

  1. UV-visible absorption spectra should be provided (at least for DDP@ZIF-8@PDA and DDP-ADL@ZIF-8@PDA). Why was an excitation wavelength of 808 nm used to evaluate the photothermal properties of the nanohybrids ?

Reply: Thank you very much for your suggestions and questions. In the previous manuscript, we mistakenly wrote FTIR as UV (Figure 2 legend), and we sincerely apologize for this error. In the revised manuscript, we have corrected “UV” to “FTIR.” Since this study did not perform UV scanning tests on the nanomedicines, we are unable to provide UV test spectra. We sincerely apologize for any inconvenience this may cause. Furthermore, the 808 nm excitation wavelength was chosen because it allows for more accurate measurement of the photothermal conversion efficiency of the nanohybrids themselves, without interference from thermal effects caused by water molecule absorption. Therefore, we selected the 808 nm excitation wavelength to evaluate the photothermal properties of the nanohybrid. This wavelength has been employed in multiple photothermal therapy studies, such as: DOI: 10.1186/s12951-020-00640-3 and DOI: 10.2147/IJN.S221496.

  1. what is the initial DDP charge in the nanoparticles ? the release kinetics of DDP as well as ADL should be studied and the results discussed in the context of literature. The advances made should be highlighted.

Reply: Thank you very much for your valuable suggestions. In the revised manuscript, we have incorporated your recommendations by adding the encapsulation efficiency and drug loading efficiency of DDP and ADL, and discussed these findings in conjunction with the literature. The specific additions are as follows:

The encapsulation rate and drug loading efficiency test results are shown in Figure S1A-B. The encapsulation efficiency of DDP in DDP@ZIF-8@PDA and DDP-ADL@ZIF-8@PDA were 96.13 ± 2.33% and 91.09 ± 1.43%, respectively, while the drug loading efficiencies were 11.33 ± 0.28% and 10.07 ± 0.28%, respectively. In formulation DDP-ADL@ZIF-8@PDA, ADL exhibited encapsulation and drug loading efficiencies of 92.43 ± 0.85% and 10.21 ± 0.38%, respectively. Compared to previous PDA drug carrier studies [17], the encapsulation efficiency of the DDP-ADL@ZIF-8@PDA prepared in this study was slightly lower (98.9% vs. 91.09–92.43%), but the drug loading capacity was relatively higher (5.7% vs. 10.07–10.21%). This discrepancy may be attributed to differences in the amount of drug loaded or variations in the preparation methods employed.

Figure S1. Drug encapsulation efficiency and loading efficiency testing. A. DDP and ADL encapsulation efficiency test; B. DDP and ADL loading efficiency testing.

Round 2

Reviewer 1 Report

Comments and Suggestions for Authors

The authors have made the necessary corrections, so I believe the manuscript can be recommended for publication in Pharmaceuticals.

Reviewer 2 Report

Comments and Suggestions for Authors

Most of my comments were considered by the authors. 

The manuscript can be accepted by Pharmaceuticals.